# A Bayesian non-parametric mixed-effects model of microbial growth curves

**Peter D. Tonner**[1,2¤], **Cynthia L. Darnell**[2], **Francesca M. L. Bushell**[3], **Peter A. Lund**[3], **Amy K. Schmid**[1,2,4]*, **Scott C. Schmidler**[1,5,6]

**1** Program in Computational Biology and Bioinformatics, Duke University, Durham, NC, USA, **2** Biology Department, Duke University, Durham, NC, USA, **3** Institute of Microbiology and Infection, School of Biosciences, University of Birmingham, Birmingham, United Kingdom, **4** Center for Computational Biology and Bioinformatics, Duke University, Durham, NC, USA, **5** Department of Statistical Science, Duke University, Durham, USA, **6** Department of Computer Science, Duke University, Durham, USA

¤ Current address: National Institute for Standards and Technology, Gaithersburg, MD, USA
* amy.schmid@duke.edu

**Data Availability Statement:** All relevant data and computer code are available for downloaded via the following GitHub repositories: https://github.com/ptonner/phenom https://github.com/ptonner/

## Abstract

Substantive changes in gene expression, metabolism, and the proteome are manifested in overall changes in microbial population growth. Quantifying how microbes grow is therefore fundamental to areas such as genetics, bioengineering, and food safety. Traditional parametric growth curve models capture the population growth behavior through a set of summarizing parameters. However, estimation of these parameters from data is confounded by random effects such as experimental variability, batch effects or differences in experimental material. A systematic statistical method to identify and correct for such confounding effects in population growth data is not currently available. Further, our previous work has demonstrated that parametric models are insufficient to explain and predict microbial response under non-standard growth conditions. Here we develop a hierarchical Bayesian non-parametric model of population growth that identifies the latent growth behavior and response to perturbation, while simultaneously correcting for random effects in the data. This model enables more accurate estimates of the biological effect of interest, while better accounting for the uncertainty due to technical variation. Additionally, modeling hierarchical variation provides estimates of the relative impact of various confounding effects on measured population growth.

## Author summary

Quantifying how microbes grow in response to stress is required for effective treatment of microbial infections, food safety, and understanding the effects of environmental change. Current models that quantify microbial growth characteristics such as exponential growth rate are based on assumptions that microbial growth curves will adopt a sigmoid form with characteristic lag, logarithmic, and stationary phases. These models are therefore inaccurate when applied to microbes growing under stress. Substantial variability across experiments that measure microbial growth further compounds the issue. Here we report a new statistical model freed from the assumption of optimum growth. This

hsalinarum_tf_phenotype https://github.com/amyschmid/pseudomonas-organic-acids.

**Funding:** This study was funded by National Science foundation grants DMS-1407622 to SCS; NSF-MCB-1615685, -1651117, and -1642283 to AKS; NSF Graduate Research Fellowship to PDT (nsf.gov). Research on this project in PAL's laboratory was supported by grant number BB/K019171/1 from the UK Biotechnology and Biological Sciences Research Council (bbsrc.ukri.org/). The funders had no role in study design, data collection and analysis, decision to publish, or preparation of the manuscript.

**Competing interests:** The authors have declared that no competing interests exist.

model also properly corrects for experimental variability, enabling researchers to monitor, quantify, and understand how microbial growth changes in response to gradations of stress. We apply this model to two microbial test systems to accurately quantify how pathogenic bacteria respond to acidic antimicrobial treatments, and how environmentally important microbes withstand stress.

This is a *PLOS Computational Biology* Methods paper.

## Introduction

Microbial growth phenotypes inform studies in microbiology, including gene functional discovery, bioengineering process development, and food safety testing [1–3]. For example, recent advances in microbial functional genomics and phenotyping, or "phenomics", have enabled transformative insights into gene functions, proving critical for mapping the genotype to phenotype relationship [4]. Methods such as genome-wide CRISPRi [5] and targeted genome-scale deletion libraries [6, 7] frequently rely upon accurate quantitation of microbial population growth as an assay to identify novel mutants with significant growth phenotypes. Population growth, as measured by the growth curve of a microbial culture, is an aggregate measure of all cellular processes and captures how microbial cells adapt and survive in their environmental niche [8]. Because microbial culturing is a necessary precursor to many experimental procedures in microbiology [9], reproducible results require accurate quantification of the variability in culture state measured through growth [9, 10].

Typical analyses of microbial population growth involve estimating parametric models under the assumptions of standard growth conditions comprised of three successive growth phases: (1) lag phase, in which the population adapts to a new environment, typically fresh growth medium at culture inoculation; (2) log phase, when the population grows exponentially at a rate dependent on nutrients in the environment; and (3) stationary phase, where measurable population growth terminates thereby reaching the culture carrying capacity [11]. Recent studies have shown that the estimates of parameters in these models are highly uncertain [12–14]. This uncertainty arises both from factors of biological interest, such as differences in genetic background and environment, as well as uncontrolled technical noise from experimental manipulation of microbial cultures. While such sources of variability can be modeled using fixed and random effects [15–19], parametric population growth models have additional limitations. Specifically, the parametric assumptions of these growth models require that growth measurements match the sigmoid shape expected for the growth curve under optimum conditions. When population growth deviates from the standard sigmoid shape assumed in these models, model extensions must be developed on a case by case basis for each new experimental perturbation [20, 21]. Additionally, we have shown in previous work that in cases such as extreme stress or strongly deleterious mutations, no parametric growth model accurately represents the growth curve, regardless of extension [19, 22, 23].

Factors affecting microbial growth measurements include both fixed and random effects [24]. Fixed effects are assumed to be drawn from a finite set of perturbations of interest, for example the effect of different concentrations of a chemical on growth that are entirely represented in the dataset. Random effects, conversely, can be viewed as a random sample from

a larger set of interest. For example, repeating the same design over many experiments corresponds to sampling the random experimental effect from the theoretical set of all possible experiments that could be conducted with this design [3, 25]. Random effects arising from repeated experimental design are typically referred to as *batch effects* [26, 27]. Batch effects are often a significant component of measurement noise in high-throughput genomics experiments [28]. However, random effects are not always due to experimental noise, and may represent quantities of direct scientific interest; for example, assaying a set of genetic backgrounds may be viewed as sampling from the set of all possible genetic variants [29–33]. Models which include both fixed and random effects are referred to as mixed effects models.

In this study we present *phenom*, a general model for analysis of phenomic growth curve experiments based on a Bayesian non-parametric functional mixed effects model of microbial growth. We demonstrate the utility of *phenom* to analyze population growth measurements of two microorganisms: the hypersaline adapted archaeon, *Halobacterium salinarum*; and the opportunistic bacterial pathogen, *Pseudomonas aeruginosa*. *H. salinarum* is a model organism for transcriptional regulation of stress response in the domain of life Archaea [34–36]. *H. salinarum* is particularly well adapted to resisting oxidative stress (OS), which arises from the buildup of reactive oxygen species and causes damage to many critical cellular components, including DNA, protein, and lipids [37–43]. Population growth measurements of *H. salinarum* under OS have been used previously to quantify these harmful effects on physiology, as well as identify regulatory factors important for OS survival [22, 40–42]. The presence of batch effects in *H. salinarum* OS response was reported (and corrected for) previously [19], but these efforts did not include modeling of individual batch effects for each term in the model. This motivated the explicit deconstruction of batch effects between different factors (e.g. strain and stress), which we have implemented in *phenom* and reported here.

*Pseudomonas aeruginosa* is an opportunistic microbial pathogen and a growing problem in hospital-borne infections. Rising antimicrobial resistance of these organisms has necessitated the development of alternative treatment strategies. For example, topical treatment of infected burn wounds with acetic or organic acids (OAs) has been successful [44]. OA impact on growth depends on external pH levels—in acidic environments the OA does not dissociate, but rather freely traverses the cellular membrane as an uncharged particle. Within the neutral cytoplasm, the OA dissociates, and the protons released induce acid stress [45]. Here we apply *phenom* to the *P. aeruginosa* dataset, which is foundational for a larger study of *P. aeruginosa* strains responding to pH and OA perturbation as a potential novel treatment of pathogenic bacterial infections [23].

Stress occurs constantly in the environment: as conditions change, mild to severe cellular damage occurs, and cells must regulate their molecular components to survive [46–49]. Population growth measurements are particularly vital to the study of stress response by providing a quantitative measure of growth differences against a non-stressed control [1]. Our model recovers fixed effects due to high and low levels of OS in *H. salinarum* and interactions between organic acid concentration and pH in *P. aeruginosa*. Random effects from multiple sources are corrected, thus enabling more accurate estimates of the biological significance of the stress treatment effect. Notably, in cases where random effect and fixed effect sizes are comparable, we demonstrate that mixed modeling is critical for accurate quantification of model uncertainty. If random effects are not included in the model, the significance of the effect of stress treatments on population growth can be erroneously overestimated. We discuss the implications of these findings for multiple areas of microbiology research.

## Materials and methods

### Experimental growth data

*H. salinarum* growth was performed as described previously [22]. Briefly, starter cultures of *H. salinarum* NRC-1 Δ*ura*3 control strain [50] were recovered from frozen stock and streaked on solid medium for single colonies. Four individual colonies per strain were grown at 42˚C with shaking at 225 r.p.m. to an optical density at 600 nm ($OD_{600}$) ∼1.8–2.0 in 3 mL of Complete Medium (CM; 250 NaCl, 20 g/l MgSO4•7H2O, 3 g/l sodium citrate, 2 g/l KCl, 10 g/l peptone) supplemented with uracil (50 $\mu$g/ml). These starter cultures were diluted to $OD_{600}$ ∼0.05 in 200 $\mu$l CM ("biological replicates") then transferred in triplicate ("technical replicates") into individual wells of a microplate. Cultures were grown in a high throughput microplate reader (Bioscreen C, Growth Curves USA, Piscataway, NJ), and culture density was monitored automatically by $OD_{600}$ every 30 minutes for 48 hours at 42˚C. High and low levels of OS were induced by adding 0.333 mM and 0.083 mM of paraquat to the media, respectively, at culture inoculation.

For *P. aeruginosa*, laboratory strain PAO1 (ATCC 15692) was grown as described in reference [23]. Briefly, cultures were grown in M9 minimal media supplemented with 0.4% (w/v) glucose and 0.2% (w/v) casamino acids and buffered with 100 mM each of MES and MOPS buffers. Initial cultures were diluted to a starting OD of 0.05 before growth in a microplate reader at a total volume of 200 $\mu$l per well. Population growth was measured with a CLARIOstar automated microplate reader (BMG Labtech) at 37˚C with 300 rpm continuous shaking. The $OD_{600}$ was recorded automatically every 15 minutes for a total of 24 hours. A full factorial design of pH and OA concentration was performed for benzoate, citric acid, and malic acid. An experimental batch corresponded to two repetitions of the experiment on separate days with a minimum of three biological replicates of each condition on each day. Two batches for each OA were performed.

All data generated or analyzed during this study are included in this published article (see github repository associated with this study, https://github.com/ptonner/phenom).

### Parametric growth curve estimation

For comparison with our non-parametric methods, parametric growth curve models were estimated using the *grofit* package in R with default parameters [51]. The logistic model was used to fit each curve. Kernel density estimates of parameter distributions were calculated with the Python scipy package with default kernel bandwidth parameters [52].

### *phenom*: A hierarchical Gaussian process model of microbial growth

**Gaussian processes.** A Gaussian process (GP) defines a non-parametric distribution over functions $f(t)$, defined by the property that any finite set of observations of $f$ follow a multivariate normal distribution [53]. A GP is fully defined by a mean function $\hat{f}(t)$ and a covariance function $\kappa(t, t')$ (Eq (1)):

$$f(t) \sim GP(\hat{f}(t), \kappa(t, t')) \tag{1}$$

GPs are commonly used for non-parametric curve fitting [53] where $\hat{f}(t)$ is typically set to 0, which we do here. Similarly, we use a common choice for covariance function defined by a radial basis function (RBF) kernel (Eq (2)).

$$\kappa(t, t') = \sigma^2 \cdot \exp\left(\frac{-|t - t'|^2}{\ell}\right) \tag{2}$$

Where $\sigma^2$ is the variance and $\ell$ is the length-scale. The parameter $\sigma^2$ controls the overall magnitude of fluctuation in the population of functions described in the GP distribution, while $\ell$ controls the expected smoothness, with larger $\ell$ making smoother, slower varying functions more likely. In the process of non-parametric modeling of growth curves, these parameters are adaptively estimated from the dataset.

**Fixed effects.** We first define the fixed effects models used in this study; these will be augmented with random effects in the next section. We consider fixed effects models of increasing complexity: a mean growth phenotype, a single treatment phenotype, and a combinatorial phenotype with interactions between treatments. All of these models fall under the functional analysis of variance (ANOVA) framework [22, 54]. To estimate a mean growth profile, as in the case of measuring a single condition, a mean function $m(t)$ is estimated from the data by modeling each replicate $y_r(t)$ for $1 \leq r \leq R$ as consisting of an unknown mean function observed with additive noise (Eq (3)).

$$y_r(t) = m(t) + \epsilon_r(t) \tag{3}$$

Where $m(t) \sim GP(0, \kappa_m(t, t'))$ provides a prior distribution over $m$, and $\kappa_m$ is an RBF kernel with hyperparameters $\{\sigma_m^2, \ell_m\}$. Here $\epsilon_r(t) \sim N(0, \sigma_y^2 I)$ is Gaussian white noise.

When estimating the effect of a perturbation on growth, as in the case of OS, we add a second function $\delta(t)$ that represents the effect of the stress being considered. The model then becomes (Eq (4)):

$$y_r(t) = \begin{cases} m(t) + \epsilon_r(t) & \text{if standard growth} \\ m(t) + \delta(t) + \epsilon_r(t) & \text{otherwise} \end{cases} \tag{4}$$

where $\delta(t) \sim GP(0, \kappa_\delta(t, t'))$ also follows a GP prior independently of $m$, and $\kappa_\delta$ has hyperparameters $\{\sigma_\delta^2, \ell_\delta\}$.

When incorporating possible interaction effects such as those between pH and organic acids in the *P. aeruginosa* dataset, the model becomes (Eq (5)):

$$y_r(t, p, m) = m(t) + \alpha_p(t) + \beta_c(t) + (\alpha\beta)_{p,c}(t) + \epsilon_r(t), \tag{5}$$

for pH $p$ and molar acid concentration $c$, with $\alpha_p(t)$ representing the main effect of pH, $\beta_c(t)$ the main effect of acid concentration, and $(\alpha\beta)_{p,c}(t)$ the interaction between them. Each effect is drawn from a treatment specific GP prior (Eq (6)).

$$\begin{aligned} \alpha_p(t) &\sim & GP(0, \kappa_\alpha(t, t')) \\ \beta_c(t) &\sim & GP(0, \kappa_\beta(t, t')) \\ (\alpha\beta)_{p,c}(t) &\sim & GP(0, \kappa_{\alpha\beta}(t, t')) \end{aligned} \tag{6}$$

Again, each covariance function is specified by a RBF kernel with corresponding variance and lengthscale hyperparameters that adapt to the observed data. All models in this section correspond to $M_{null}$ for their respective analyses, as they do not include any random effects.

**Random effects.** The first random effects added to the model were those used to account for batch effects, in the model $M_{batch}$. Under this model, each fixed functional effect is modified by a GP describing the population of possible batch-specific curves. For example, under the model of interaction effects on growth (Eq 5), replicate $r$ from batch $k$ is modeled as

(Eq 7)):

$$M_{\text{batch}}: \quad y_{k,r}(t,p,m) = \overbrace{m(t) + \alpha_p(t) + \beta_c(t) + (\alpha\beta)_{p,c}(t)}^{M_{\text{null}}}$$
$$+ m^{(k)}(t) + \alpha_p^{(k)}(t) + (\alpha\beta)_{p,c}^{(k)}(t) + \beta_c^{(k)}(t) + \epsilon_{k,r}(t) \qquad (7)$$

where the functions shared with $M_{\text{null}}$ are highlighted, the functions $m^{(k)}(t)$, $\alpha_p^{(k)}(t)$, $\beta_c^{(k)}(t)$, and $(\alpha\beta)_{p,c}^{(k)}(t)$ are the corresponding random batch effects, and $\epsilon_{k,r}(t) \sim N(0, \sigma_y^2 I)$. Similar to the fixed effects, the batch effect functions are drawn from shared GP priors (Eq (8)):

$$\alpha_p^{(k)}(t) \sim \quad GP(0, \kappa_{\alpha,\text{batch}}(t, t'))$$
$$\beta_c^{(k)}(t) \sim \quad GP(0, \kappa_{\beta,\text{batch}}(t, t')) \qquad (8)$$
$$(\alpha\beta)_{p,c}^{(k)}(t) \sim \quad GP(0, \kappa_{\alpha\beta,\text{batch}}(t, t')),$$

with kernel hyperparameters $\{\sigma_{\alpha,\text{batch}}^2, \ell_{\alpha,\text{batch}}\}$, $\{\sigma_{\beta,\text{batch}}^2, \ell_{\beta,\text{batch}}\}$, and $\{\sigma_{\alpha\beta,\text{batch}}^2, \ell_{\alpha\beta,\text{batch}}\}$ that are distinct from those for the corresponding fixed effects, allowing for different variance and lengthscales between fixed and random effects. Other $M_{\text{null}}$ models are converted to $M_{\text{batch}}$ similarly, with each fixed effect becoming a mean of a GP prior for each batch effect.

$M_{\text{full}}$ develops the hierarchy one step deeper by adding replicate effects to $M_{\text{batch}}$ (Eq 7). Specifically, the error model $\epsilon_{k,r}$ is now described by a GP: $\epsilon_{k,r} \sim GP(0, \kappa_y(t, t'))$ with corresponding hyperparameters, accounting for replicate-specific variability rather than simply white noise.

**Bayesian inference.** The unknown functions ($m(t)$, $\delta(t)$, $\alpha_p(t)$, $\beta_c(t)$, and $(\alpha\beta)_{p,c}(t)$), kernel hyperparameters ($\sigma_l^2$ and $\ell_l$) for each group of latent functions, and observation noise parameters ($\sigma_y^2$) are all estimated by Bayesian statistical inference. In Bayesian inference, prior distributions on unknown quantities (e.g. $p(m(t), \theta_m) = p(m(t)|\theta_m) \times p(\theta_m)$) are combined with the likelihood, $p(y(t)|m(t))$ to obtain the posterior distribution $p(m(t), \theta_m|y(t))$.

Latent functions are grouped by shared kernel hyperparameters $\theta_l = \{\sigma_l^2, \ell_l\}$ into related sets (e.g. treatment effects, interaction effects, batch effects), which then provide the GP prior for the latent function. For each group, $\sigma_l^2$ is assigned a Gamma($\alpha$, $\beta$) prior, with fixed effects assigned as a Gamma(10, 10) prior and random effects assigned a Gamma(7, 10) prior. $\ell_l$ was assigned an inverse- Gamma($\alpha$, $\beta$) prior, with parameter $\alpha = 6$ and $\beta = 1$ for *H. salinarum*, and $\alpha = 2$, $\beta = 3$ for *P. aeruginosa* fixed effects and $\alpha = 10$, $\beta = 1$ for *P. aeruginosa* random effects. Noise variance $\sigma_y^2$ was also assigned a gamma prior.

Bayesian inference was then performed, with the posterior distribution obtained by sampling using Markov chain Monte Carlo (MCMC) implemented with the Stan library, which uses a Hamilitonian Monte-Carlo procedure with No-U-turn sampling [55]. Multiple chains were run to diagnose convergence, with all parameter posterior means confirmed to have converged within $\hat{R} < 1.1$ as recommended [56].

## Results

### Hierarchical batch effects typical in phenomics datasets render parametric models ineffective

In the dataset used here, population growth for each of *P. aeruginosa* and *H. salinarum* cultures was monitored under standard (non-stressed) conditions vs. stress conditions (see Materials and methods and references [22, 23] for precise definition of "standard conditions"

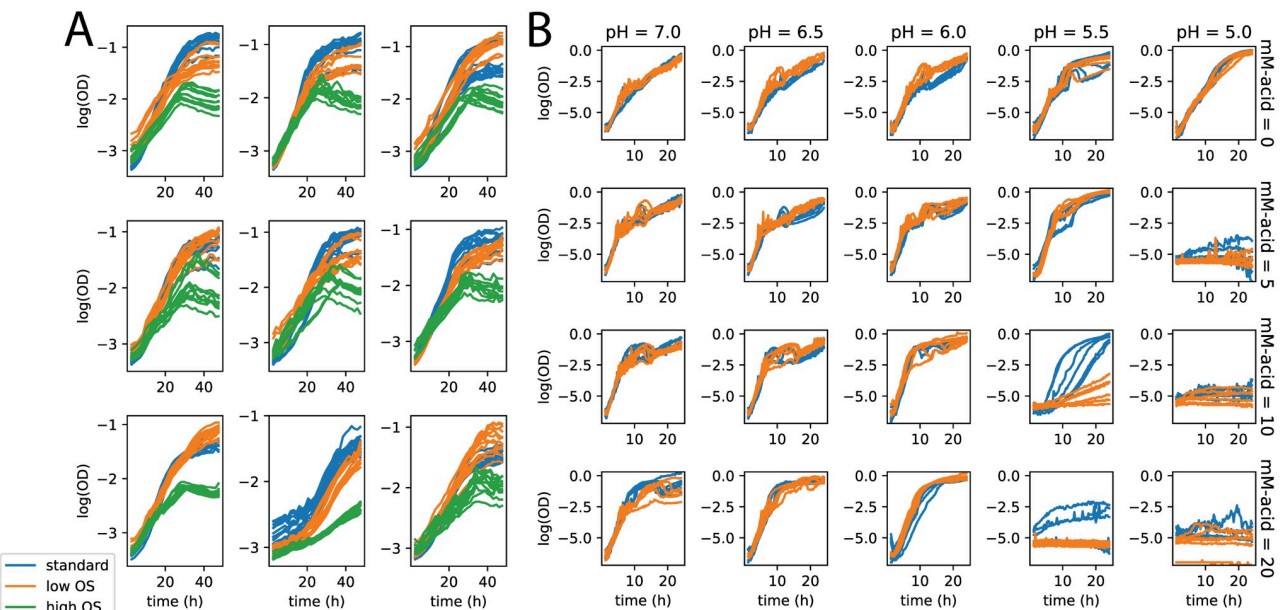

**Fig 1. Batch variation in high throughput phenomics studies.** A: Population growth measurements of *H. salinarum* under standard conditions (blue), and low (orange) and high (green) levels of OS. Individual measurement curves are replicates and each graph panel is a different batch. B: Growth of *P. aeruginosa* strain PA01 under gradient of pH (5–7) and citric acid (0–20 mM). Colors represent different batches.

for each organism). Specifically, cultures were grown in liquid medium in a high throughput growth plate reader that measured population density at 30 minute intervals over the course of 24 hours (*P. aeruginosa*) or 48 hours (*H. salinarum*); the resulting data are shown in Fig 1. Experimental designs for each organism included biological replicates (growth curves from different colonies on a plate), technical replicates (multiple growth curves from the same colony), varying conditions (stress vs standard), and batches. Throughout, we define "batch" as a single run of the high throughput growth plate reader. In each run, this plate reader measures the growth of 200 individual cultures across a range of perturbations, including varying stress conditions and genetic mutations (see Methods). *H. salinarum* was grown under high (0.333 mM paraquat (PQ)) and low (0.083 mM PQ) levels of oxidative stress (OS); the data are combined from published [19, 22, 41] and unpublished studies (Fig 1A). The OS responses of *H. salinarum* were compared to a control of standard growth in rich medium, representing optimal conditions for the population. The experimental design was replicated in biological quadruplicate and technical triplicate, across nine batches (Fig 1A, individual curves and axes). *P. aeruginosa* was grown in the presence of increasing concentrations of three different organic acid (OA) chemicals (0–20mM; benzoate, citric acid, and malic acid), each combined with a gradient of pH (5.0–7.0) [23]. Each *P. aeruginosa* growth condition was repeated across 3 biological replicates and two batches (Fig 1B). The different *P. aeruginosa* and *H. salinarum* experimental designs with varying numbers of replicates at each level provides a rich test bed for modeling the impact of random effects with *phenom* (Fig 1B, S1 and S2 Figs).

Figs 1 and 2 demonstrate the two key issues described above and addressed in this paper. First, batch effects are present in both *H. salinarum* and the *P. aeruginosa* datasets. For *H. salinarum*, clear differences in growth under both standard and stress conditions are observed in the raw data across experimental batches (i.e. separate runs of the growth plate reader instrument; Fig 1). Some batches show a different phenotype, with either a complete cessation of growth or an intermediate effect with decreased growth relative to standard conditions. For

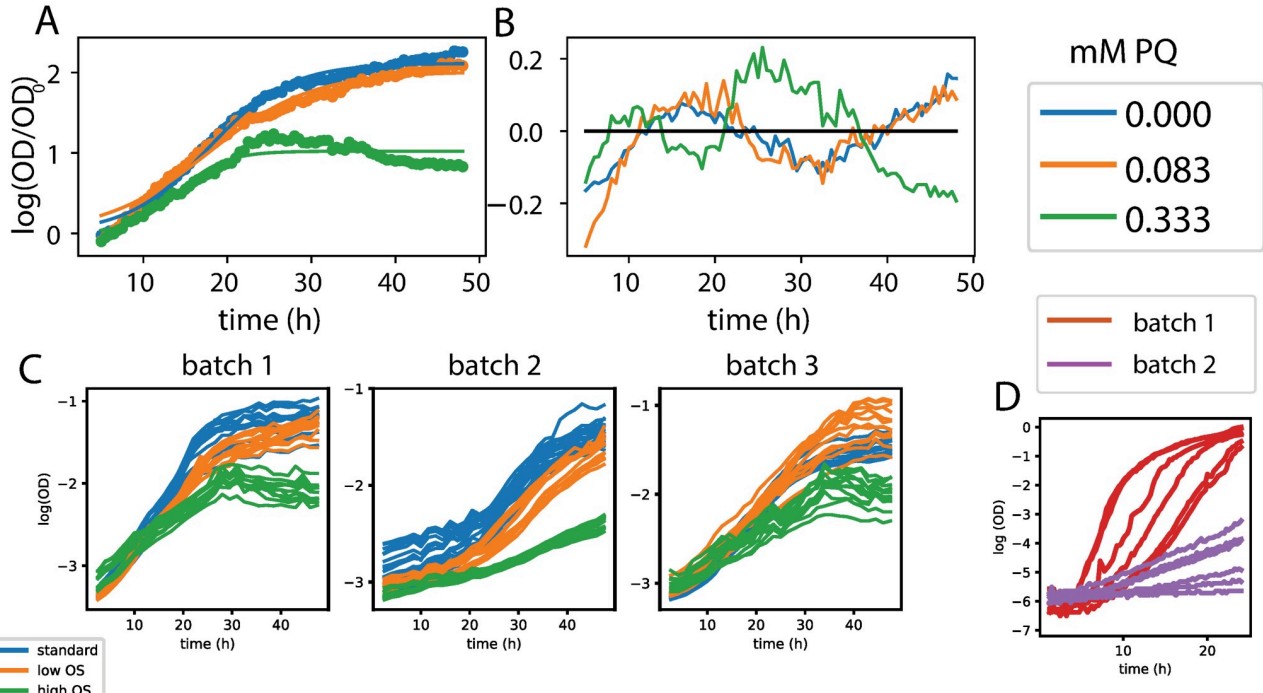

**Fig 2. Batch effects are prevalent in microbial phenomic datasets.** (A) Parametric fits to *H. salinarum* growth curves. (B) Residuals of parametric growth curve fit. (C) Growth of *H. salinarum* under standard conditions (blue), low (orange) and high (green) OS across three batches. (D) Measurement of *P. aeruginosa* growth under 10mM citric acid at 5.5 pH. Measurements for each condition vary significantly with batch.

example, in some batches, populations stressed with low OS grow at the same rate and reach the same carrying capacity as populations grown under standard conditions. For *P. aeruginosa*, a clear difference between batches grown under 10 mM citric acid at pH = 5.5 is observed [Fig 1B (graph in fourth column, third row) and Fig 2D]. Like with citric acid, batch effects were also found in some of the other conditions considered (e.g. growth under malic acid, S1 and S2 Figs).

Second, standard parametric growth curve models fail to describe experimental measurements adequately (Fig 2A and 2B), as we have shown previously with both datasets [19, 22, 23]. In Fig 2, we examined the impact of batch and replicate effects on our data by considering how they change parameters estimated from a mixed effects parametric model of population growth [32]. We focused on calculating $\mu_{max}$, the maximum instantaneous growth rate attained by the population, as this is a commonly used parameter for comparisons between conditions [19, 57]. Variation in $\mu_{max}$ estimates were observed both on the replicate and batch level, as shown by the kernel density estimates (KDE) of $\mu_{max}$ for each stress level (S3 Fig). The variance in $\mu_{max}$ is remarkably high: the 95% confidence interval for $\mu_{max}$ under standard growth is 0.050–0.141, a nearly 3-fold change between the lower and upper interval limits. Thus, while the t-test conducted on $\mu_{max}$ estimates between standard conditions and each stress level is statistically significant (S3 Fig), it is difficult to conclude: (a) what the true magnitude of the stress effects may be; and (b) to what degree the variation due to replicate and batch should inform biological conclusions. The error of the logistic growth model under each PQ condition was also examined. Error increased under high OS (S4 Fig). High OS induces a growth phenotype that deviates heavily from the sigmoidal growth curve assumed in the logistic model as well as in other commonly used growth models. This leads to a poor fit under the

high OS condition as has been shown previously (S4 Fig, [19]). The residuals under standard, low, and high OS conditions also appear to be dependent. Our previous work also demonstrated poor fits to the *P. aeruginosa* data using parametric models [23]. Taken together, the initial assessment of these two datasets indicates that: (a) technical variation due to batch and replicate in growth curve data can be high; and (b) commonly used standard parametric models are not able to adequately capture or correct for these sources of variability. These sources of error need to be corrected in order to model true growth behavior and inform biological conclusions from the data.

## A hierarchical Bayesian model of functional random effects in microbial growth

We previously established the ability of non-parametric Bayesian methods to improve the modeling of growth phenotypes [19, 22, 23]. Here, we describe *phenom*, a fully hierarchical Bayesian non-parametric functional mixed effects model for population growth data. We highlight the utility of *phenom* to correct for confounding, random effects in growth phenotypes.

In order to model both biological and technical variation in microbial growth (Fig 3), we first assume that a set of population growth measurements are driven by an (unobserved) population curve $m(t)$ (Fig 3A, blue curve) of unknown shape. For example, $m(t)$ might represent the average growth behavior of an organism under standard conditions. This mean growth behavior may be altered by a treatment effect, represented by an additional unknown curve $\delta(t)$ (Fig 3A, orange curve). For example $\delta(t)$ may represent the effects on growth induced by low or high levels of OS (Fig 2A). The average growth behavior of a population under stress conditions would then be described by the curve $f(t) = m(t) + \delta(t)$.

When considering a combinatorial experimental design, such as that described for *P. aeruginosa* growth (Fig 1B), we model independent effects of different treatments as well as their interaction via the form (Eq (9)):

$$y(t, i, j) = m(t) + \alpha_i(t) + \beta_j(t) + (\alpha\beta)_{i,j}(t). \tag{9}$$

Here, $y(t, i, j)$ denotes the observed population size at time $t$ with treatments $i$ and $j$ of two independent stress conditions. Additionally, $\alpha_i(t)$ and $\beta_j(t)$ are the independent effects of each stress condition, and $(\alpha\beta)_{i,j}(t)$ is their interaction. This model corresponds to a functional analysis of variance [58], which we have previously used to estimate independent and interaction effects of microbial genetics and stress [22]. For the analysis of *P. aeruginosa*, we model the effect of pH ($\alpha_p$ for pH = p), organic acid combination ($\beta_c$ for concentration = c) and their interaction ($(\alpha\beta)_{p,c}$), as well as their random functional effect equivalents (see Section "*phenom*: A hierarchical Gaussian process model of microbial growth").

Variability around these fixed effect growth models is described by additional, random curves associated with two major sources of variation: *batch* and *replicate* (Fig 3B and 3C). Batches correspond to a single high-throughput growth experiment and replicates are the individual curve observations within a batch. Using *phenom* throughout this study, we only compare replicates that are contained within the same batch. This is due to the nested structure between batch and replicates (Fig 3). Noise due to both replicate and batch do not appear to be independent identically distributed (*iid*), as observed in the correlated residuals around the mean for each experimental variate (S5A and S5B Fig). Each observed growth curve is therefore described by a combination of the fixed effects and the corresponding batch and replicate effects (Fig 3D). Both replicate and batch variation are modeled as random effects because the variation due to both sources cannot be replicated, i.e. a specific batch effect cannot be purposefully re-introduced in subsequent experiments. Instead, these variates are assumed to be

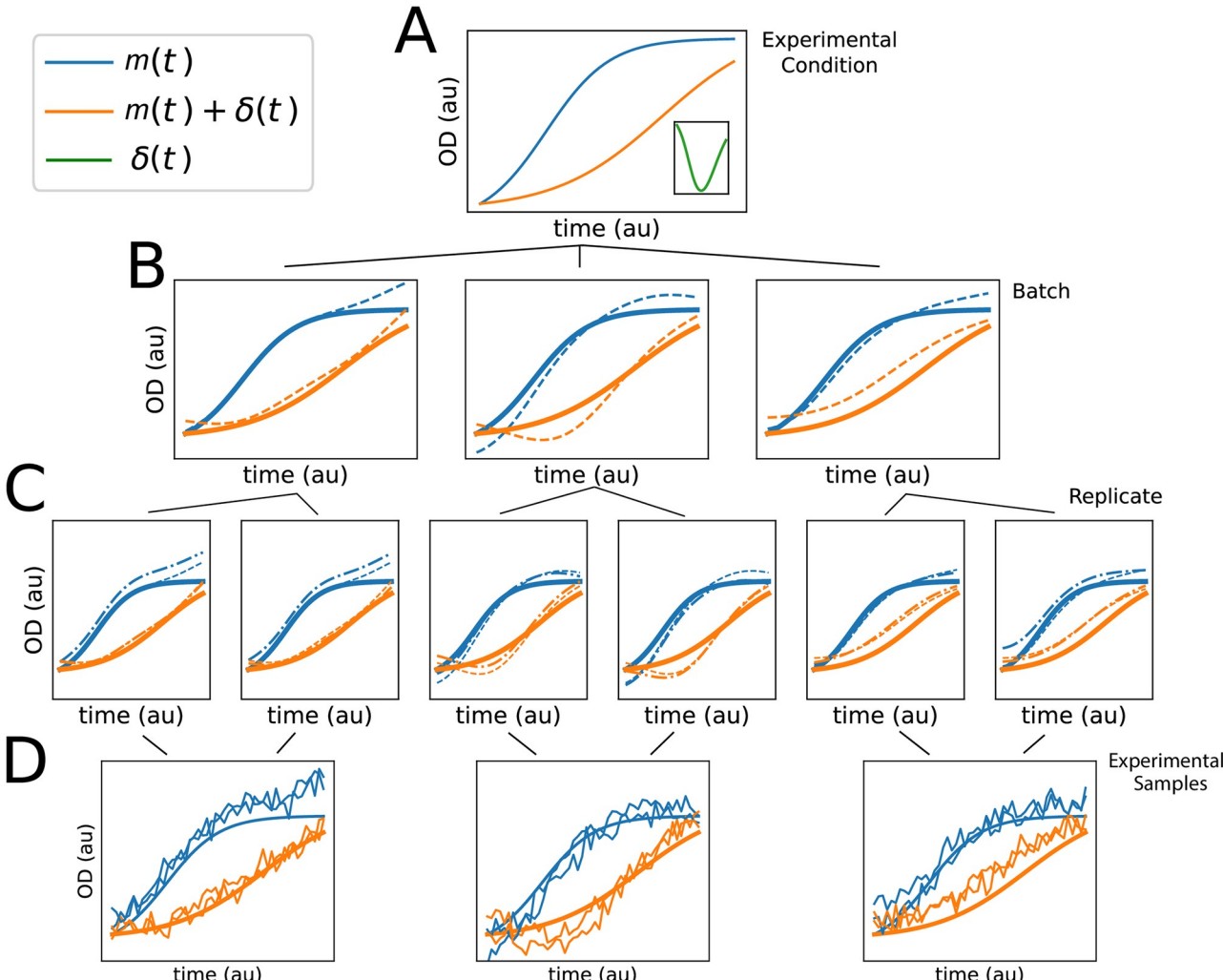

**Fig 3. Hierarchical model of functional data.** Representative diagram of hierarchical variation present in microbial growth data. Each tier of graphs represents a different variation source, and lines indicate relationship between them: experimental condition is the true growth behavior of interest, with the condition repeated across batches, and replicates repeated within each batch. (A) Functional phenotypes $m(t)$ (blue), $m(t) + \delta(t)$ (orange), and $\delta(t)$ (green curve in inset). (B) Batch effects on $m(t)$ and $m(t) + \delta(t)$. Each plot is a different batch, solid lines are the true functions as in (A), and the dashed lines are the observed batch effect of $m(t)$ and $m(t) + \delta(t)$ for the corresponding batch. (C) Replicate effect within batches. Each axis is a different replicate, solid and dashed lines as in (B), dotted-dashed line is the observed replicate function. (D) Observations from the model described in (A-C). Each curve is sampled with a mean drawn from the global mean, with added batch and replicate effects (dotted-dashed lines in C) and *iid* observation noise. Each axis is a different batch. The smooth solid lines are the true functions $m(t)$ and $m(t) + \delta(t)$ in (A).

sampled from a latent distribution [59]. Combining the fixed and random effects, we arrive at a mixed-effects model of microbial phenotypes.

We adopted a hierarchical Bayesian framework to model these mixed effects. In this framework, batch effects are described by a shared generative distribution, allowing them to take on distinct values while still pooling across replicates for accurately estimating the generating distribution [60]. We use Gaussian process (GP) distributions for all groups in the model. GPs are flexible, non-parametric distributions suitable for smooth functions [53]. To assess the impact of incorporating random effects on estimation of the treatment effect of interest, we analyze three models of increasing complexity: $M_{null}$ excludes all hierarchical random effects, $M_{batch}$ incorporates batch variation only, and $M_{full}$ incorporates both batch and replicate

variation. These models, collectively called *phenom*, were implemented using the probabilistic programming language Stan [55] to perform Bayesian statistical inference for all unknown functions and model parameters through Hamiltonian Monte Carlo sampling (see Materials and methods).

In previous work [19] we identified and corrected for batch effects in a single transcription factor mutant's stress response, but this model did not provide an explicit deconstruction of batch effects between different factors (e.g. strain and stress) and could therefore not determine which factors were most strongly impacted by batch effects. Moreover, this approach utilized a standard GP regression framework, which has well-established limitations on dataset size, limiting its applicability to the large datasets we consider here. In reference [22] we described a functional ANOVA model for microbial growth phenotypes, which corresponds to the $M_{null}$ model in the *phenom* case. Again, a global batch effects term was included but individual batch effects were not modeled, and the computational approach utilized (Gibbs sampling) was prohibitively slow for the complete *phenom* model. *phenom* represents a significant advance on these previous modeling approaches and computational methodologies.

In order to demonstrate the impact of batch effects on the conclusions drawn from the analysis of microbial growth data, we estimated the latent functions driving both *H. salinarum* and *P. aeruginosa* growth using the $M_{null}$ model of *phenom*, with each batch analyzed separately (Fig 4). This corresponds to the analysis that would be conducted after generating any single set of experiments from a batch, without considering or controlling for batch effects, and therefore provides a test of the impact of ignoring batch effects.

For *H. salinarum*, growth data under standard conditions was used to estimate a single mean function, $m(t)$, and fixed effects were estimated for differential growth under low and high OS as $\delta(t)$ (Fig 4A). For the *P. aeruginosa* dataset, batch effects on the interaction between pH and organic acid concentration was represented by a function $(\alpha\beta)_{p,c}(t)$, again estimated non-parametrically (Fig 4B). However, rather than reporting $(\alpha\beta)_{p,c}(t)$ directly, we report its time derivative, which has the interpretation of instantaneous growth rate rather than absolute amount of growth, and provides an alternative metric for assessing the significance of a treatment effect on growth [61]. Specifically, assessing growth curve models can benefit from the estimates of derivatives as they may more accurately represent the differences between growth curves [58].

Fitting the $M_{null}$ model to each separate batch reveals that the posterior distributions obtained for each function of interest ($m(t)$, $\delta(t)$, and $(\alpha\beta)_{p,c}(t)$) are highly variable across batches (Fig 4). This is observed in both the *H. salinarum* and *P. aeruginosa* datasets, where the experimental conditions, and therefore the underlying true mean functions, remain constant across batches in each case. Such variability can impact conclusions. We specifically assess the changes in statistically significant treatment effects, i.e. at time points where the effect ($\delta(t)$ or $(\alpha\beta)_{p,c}(t)$) has a 95% posterior credible interval excluding zero, indicating high confidence that the treatment effect at that time-point differs from the control. For example, in the low OS condition in the *H. salinarum* dataset, both the statistical significance of $\delta(t)$ and the sign (improved vs. impaired growth) differs between batches (Fig 4A, center). Additionally, the effect of low oxidative stress at time zero is estimated to be non-zero for many of the batches. This is due to technical variation that introduces an artificial offset in OD measurements at the beginning of the growth experiment. Such variation can arise from various factors, including variation between growth state in starter cultures and technical variation in plate reader measurements at low OD (Fig 1A). A similar batch variability was observed under high OS, but due to the stronger effect of the stress perturbation, estimates of $\delta(t)$ are less affected by batch and replicate variation (Fig 4A, right). Similarly, the batch variability observed in the raw *P. aeruginosa* growth data (Fig 1B) results in significantly different

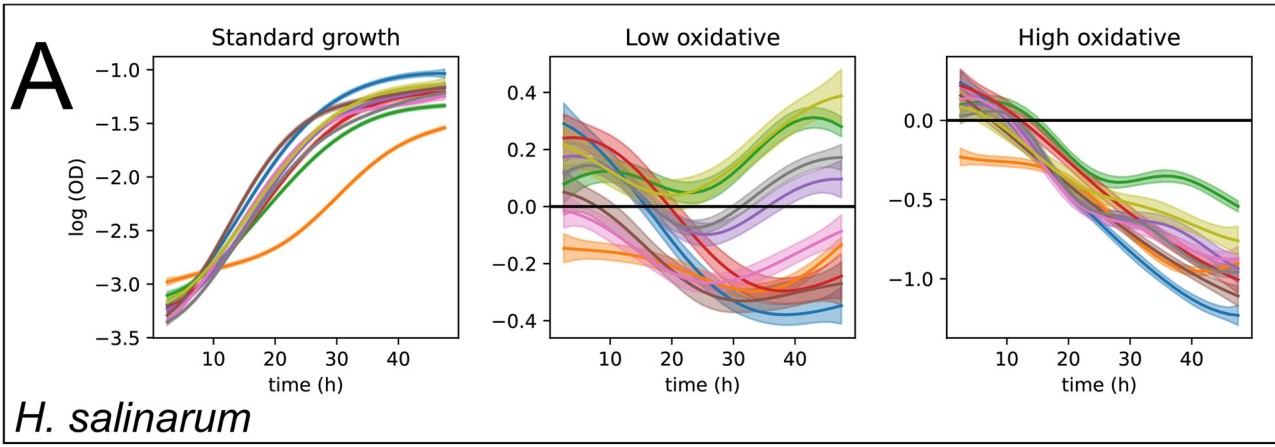

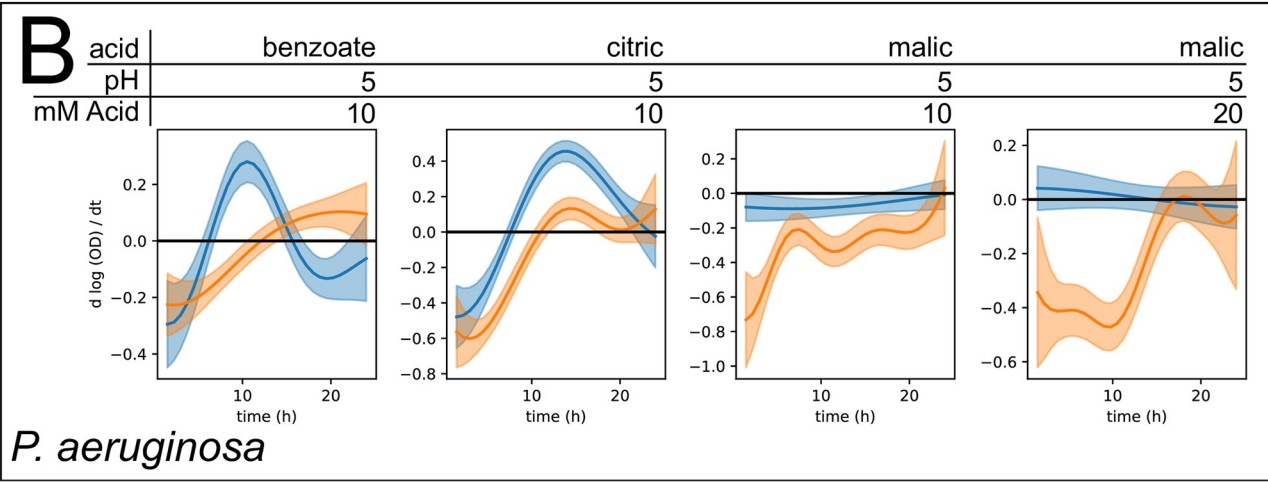

**Fig 4. $M_{null}$ model estimates are confounded by batch effects.** Posterior intervals of functions are shown for different analyses where *phenom* $M_{null}$ was fit using data from each batch separately. In all plots, solid line represents posterior mean, shaded region indicates 95% credible region, and each color corresponds to a different posterior conditioned on data from a single batch. (A) Posterior intervals of $m(t)$ under standard growth, and $\delta(t)$ under low and high OS response for *H. salinarum*. (B) Posterior interval of interaction function $(\alpha\beta)_{p,c}(t)$ for *P. aeruginosa* growth in indicated pH and acid concentration.

posterior estimates of the interaction effect $(\alpha\beta)_{p,c}(t)$ across batches, as seen by the lack of overlap between 95% credible intervals (Fig 4B). Differences observed include the timing and length of negative growth impact (benzoate and citric acid), and completely opposite effects with either strong or no interaction (malic acid). In addition, the posterior variance of each function, which indicates the level of uncertainty remaining, is low for each batch modeled separately. This indicates high confidence in the estimated function despite observed differences across batches. These analyses suggest that use of a single experimental batch leads to overconfidence in explaining the true underlying growth behavior.

## Hierarchical models correct for batch effects in growth data

To demonstrate the use of *phenom* to combat the impact of batch effects on growth curve analysis, we combined data across all batches and performed the analysis using each of the $M_{null}$, $M_{batch}$, and $M_{full}$ models (Fig 5). Estimates of $m(t)$ between each model were largely similar, likely due to the abundance of data present to estimate this variable (S6 Fig). Instead, we focus on the estimates of $\delta(t)$ for low and high OS response of *H. salinarum* (Fig 5A) and the

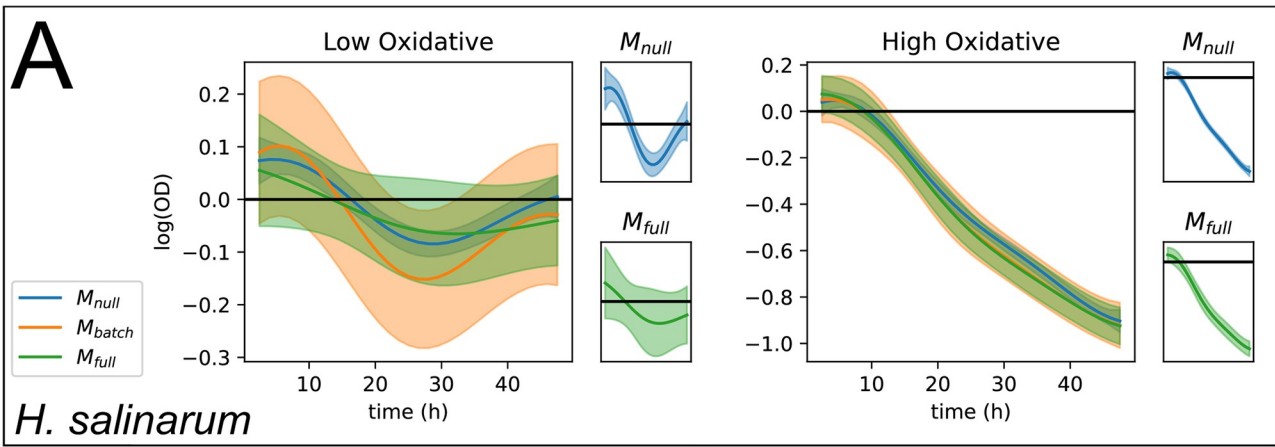

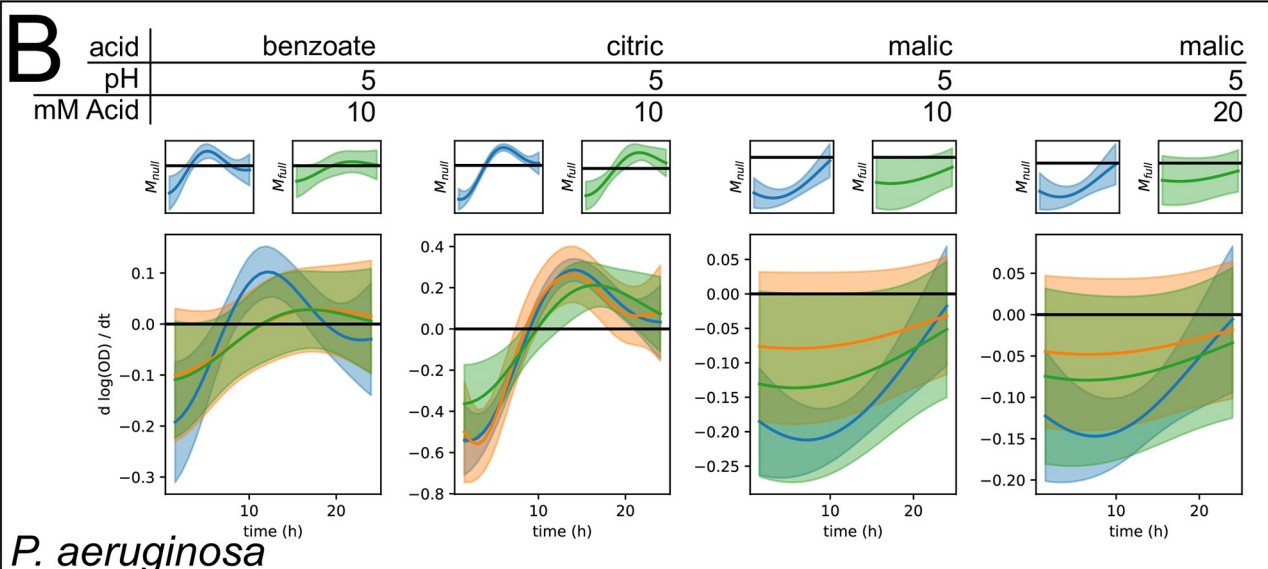

**Fig 5. Hierarchical models of growth control for batch effects.** Posterior intervals of functions estimated by models of increasing hierarchical complexity: $M_{null}$ (blue), $M_{batch}$ (orange), and $M_{full}$ (green). Solid line indicates posterior mean and shaded regions indicate 95% credible regions. We also plot $M_{null}$ and $M_{full}$ in separate axes to highlight the posterior intervals estimated by these models. (A) Posterior interval of $\delta(t)$ for low (left) and high (right) OS response by *H. salinarum*. (B) Posterior interval of interaction function $(\alpha\beta)_{p,c}(t)$ for *P. aeruginosa* growth in indicated pH and organic acid concentration.

interaction $(\alpha\beta)_{p,c}$ between pH and OA concentration effects on *P. aeruginosa* growth (Fig 5B). In cases where $M_{null}$ differs from $M_{batch}$ and $M_{full}$, this indicates an inability for this model to correctly represent the uncertainty due to random effects in the data, which have been shown to be prevalent across different batches of experiments (Fig 4).

Growth impairment in the presence of low OS relative to standard conditions (i.e. $\delta(t)$) is estimated to be significant during the time points of ∼0–13 and ∼19–40 hours under $M_{null}$. In contrast, only time points ∼23–31 are significantly non-zero under $M_{batch}$, and no significant effect is identified under $M_{full}$(Fig 5A, left). Conversely, due to the stronger stress effect in the high OS condition (Fig 5A, right), estimates of $\delta(t)$ were significantly non-zero under all three models, with only minor differences between the three model estimates. This highlights the importance of controlling for batch and replicate variability as in $M_{full}$: even when estimating the low OS treatment effect under $M_{null}$ with all available data, without accounting for batch

and replicate random effects the posterior estimates of $\delta(t)$ are overconfident and do not accurately represent the uncertainty with respect to the true treatment effect. The lack of significance under $M_{full}$ suggests that additional data are needed to confidently identify the true treatment effect in the presence of batch and replicate variation. Modeling the batch effects has also corrected for variability in treatment effects due to technical variation at the inoculation of the growth plate (time zero of the experiments) (S7 Fig).

The impact of modeling hierarchical variation on estimating interaction effects in *P. aeruginosa* growth was condition dependent (Fig 5B). Across all conditions, a decrease in posterior certainty on the true shape of the underlying function was again observed under $M_{batch}$ and $M_{full}$. For benzoate and malic acid, the interaction between pH and acid concentration no longer appears to be a significant effect after accounting for batch and replicate variation, while the larger interaction under citric acid remains significant. As in the comparison of oxidative stress treatments in *H. salinarum*, stronger effect sizes are required to be confidently distinguished in the face of batch and replicate variability. Finally, the relative conclusions made for the absolute function scale are comparable to those of the derivative estimates for *P. aeruginosa*, highlighting the flexibility with which treatment effects can be analyzed as most relevant to the researcher (S8 Fig).

For both *H. salinarum* response to OS and *P. aeruginosa* growth under pH and OA exposure, an increase in posterior variance was observed under $M_{batch}$ and $M_{full}$ compared to $M_{null}$ (S9 Fig). However, posterior variance of $\delta(t)$ in the *H. salinarum* OS response was higher under $M_{batch}$ compared to $M_{full}$. In this case, controlling for replicate effects appears to increase the signal needed to identify $\delta(t)$. In contrast, these variances are equal in the *P. aeruginosa* data, indicating that the relative improvement in variance afforded by modeling batch vs. replicate effects may be dataset dependent.

## Variance components demonstrate the importance of controlling for batch effects

Variance components, which correspond to the estimated variance of each effect in the model, can be used to compare the impact each group has on the process of interest [24]. To better understand sources of variability in growth curve studies, we used *phenom* to estimate the variance components for each dataset above. In our hierarchical non-parametric setup, these variance components are the variance hyperparameters (e.g. $\sigma^2$) of the Gaussian process kernels for each fixed and random effect group. These parameters control the magnitude of function fluctuations modeled by the GP distribution. Larger variance implies higher effect sizes and therefore a larger impact on the observations.

We show the value of variance components by considering the effects identified by $M_{full}$ for *H. salinarum* under low OS (Fig 6). The variance of the data is partitioned between the mean growth ($m(t)$), the OS ($\delta(t)$), batch effects (batch curves of $m(t)$ and $\delta(t)$), biological noise (e.g. replicate variability) and instrument noise ($\sigma_y^2$). This analysis confirms that batch effects, compared to the other sources of experimental variability in the dataset (replicate noise and measurement error), are between 2 to 10 times more impactful on the phenotype measurements. Additionally, variance components enable comparisons between the experimental and treatment factors in the data. Of particular note is that the variance of the treatment of interest, $\delta(t)$, and the batch effects are similar in magnitude, at least in the case of a low-magnitude stress such as 0.083 PQ for *H. salinarum*. This suggests that proper modeling of this treatment requires both sufficient batch replication and accurate modeling of batch effects in those data. Future studies of similar phenotypes can be guided by these estimates in experimental design, choosing an appropriate batch replication for the degree of noise expected [62]. However, the

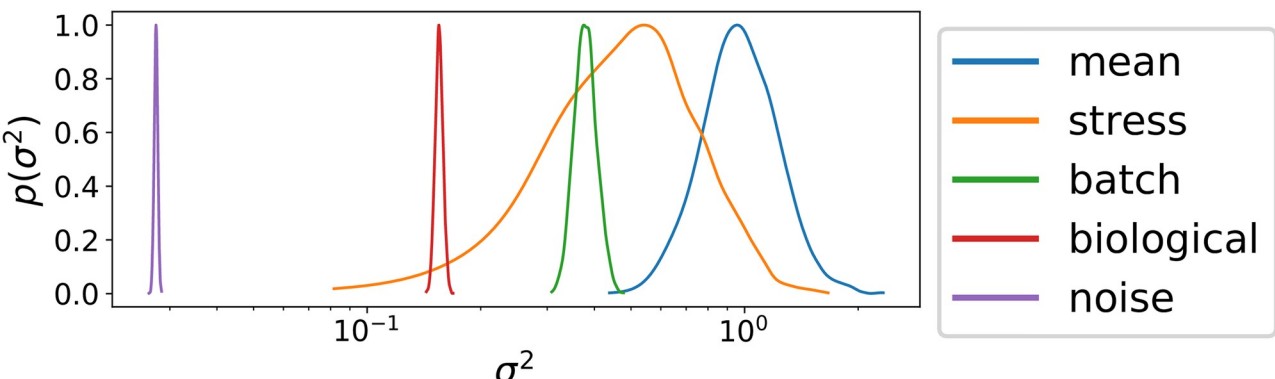

**Fig 6. Posterior variance components in the *phenom* hierarchical phenotype model.** Posterior intervals are shown for the kernel variance hyperparameter for different groups of effects from *phenom* estimated on *H. salinarum* growth under low OS. Groups correspond to $m(t)$ (mean), $\delta(t)$ (stress), batch effects (batch), replicate noise (biological), and measurement error (noise).

extent of replication required may depend upon the dataset (factorial design, treatment severity, etc). Taken together, variance components provide an aggregated view of the contribution by various factors and guide future experimentation. Iterative rounds of *phenom* analysis and growth experiments can then determine the best suited designs, for example by leveraging the estimated batch effect variance from pilot experiments to determine the number of batch replications necessary to reliably estimate a treatment effect of given magnitude. The use of *phenom* for such formal statistical experimental design calculations represents an exciting direction for future work.

## Discussion

We have provided a framework to test and control for random effects in microbial growth data using the hierarchical non-parametric Bayesian model, *phenom* (Fig 3). Analysis with *phenom* indicates that random effects (both batch and replicate) appear in the two microbial population growth datasets studied here, and constitute significant portions of the variability (Fig 1). Failure to correct for these effects confounds the interpretation of growth phenotypes for factors of interest in a large scale phenotyping analysis (Fig 4). *phenom* controls for these random effects and provides accurate estimates of the growth behavior of interest (Fig 5). Additionally, *phenom* can be used to estimate variance components, providing information about the relative impact of various sources of noise in the data (Fig 6). Controlling for batch effects in these datasets was therefore key to making accurate biological conclusions.

Related fields of functional genomics, such as transcriptomics, have seen considerable interest in controlling for different experimental sources of variation, broadly labeled as batch effects [28, 62–67]. These studies have shown that differences between batches first need to be corrected to avoid erroneous conclusions [68]. Here we have shown that, like in transcriptomics data, controlling for sources of variation in phenomics data—particularly due to batch—are an important step in making accurate biological conclusions regarding population growth. Additionally, the use of random batch effects in *phenom* highlights cases where additional information may be gained by further experimentation. Specifically, in cases where treatment effects differ strongly across batches, there may be underlying biological differences driving the variation. Follow-up experimental designs can then aim to delineate these effects directly in a way not confounded by batch. *phenom* establishes a complete and general method of controlling batch effects in microbial growth phenotypes, overcoming significant weaknesses of previously developed techniques.

Although we focus here on replicate and batch variation, the *phenom* model is easily extended to incorporate alternative or additional random and fixed effects appropriate for settings with other sources of variation. For example, depending on the experimental design, *phenom* could control for variation among labs, experimental material, culture history, or genetic background [25, 69–75]. *phenom* flexibly incorporates additional sources of variation and/or interaction between design variables, as demonstrated with the two different designs analyzed for *H. salinarum* and *P. aeruginosa* here. This flexibility allows *phenom* to be applied to control for many sources of technical variation within microbial population growth data, thereby improving the analysis and resulting conclusions regarding quantitative microbial phenotypes. We therefore expect our model to find broad applications in fields such as bioprocess control, microbial bioengineering, and microbial physiology.

## Supporting information

**S1 Fig. *P. aeruginosa* growth under benzoate and pH gradient.** Growth of *P. aeruginosa* strain PAO1 under gradient of pH (7–5) and benzoate (0–20). Colors represent different batches.
(EPS)

**S2 Fig. *P. aeruginosa* growth under malic acid and pH gradient.** Growth of *P. aeruginosa* strain PAO1 under gradient of pH (7–5) and malic acid (0–20). Colors represent different batches.
(EPS)

**S3 Fig. KDE of $\mu_{max}$ for *H. salinarum* growth across batches.** Crosses indicate significant difference between $\mu_{max}$ standard conditions and each OS level (one-sided t-test, $p < 0.05$).
(EPS)

**S4 Fig. Error in parametric growth models.** Distribution of error (MSE) for each condition when fit with a logistic growth curve. The box show shows the inter-quartile range, red line is the median, whiskers show the 1.5 inter-quartile range, and the individual points are outliers.
(EPS)

**S5 Fig. Residual structure of microbial growth data across batches.** (A) Individual replicate curve residuals around the mean of the respective batch. Only standard conditions are shown. (B) Residual of the mean behavior for each batch around the global mean (standard condition only).
(EPS)

**S6 Fig. Posterior comparison of $m(t)$ for *H. salinarum* growth across batches.** Posterior interval of $m(t)$ for *H. salinarum* standard growth.
(EPS)

**S7 Fig. Posterior intervals of low oxidative stress batch effects estimated from $M_{full}$.** Full estimates of the $\delta(t)$ batch effect under $M_{full}$ are shown, with solid lines representing posterior mean and shaded region representing 95% credible intervals (left). 95% posterior credible interval of batch effects for $\delta(t)$ at time zero are shown (right), with crosses marking posterior means. Many of the batch effects for $\delta(t)$ are estimated to be non-zero at the start of the experiment, reflecting the impact of technical variation in the high-throughput readings at the start of the growth curves.
(EPS)

**S8 Fig. Posterior intervals of interactions for *P. aeruginosa* on an absolute growth scale.** Posterior intervals of interactions in Fig 5B, but reported here on an absolute (log OD) scale. The same data is reported on the derivative (d log OD / dt) scale in the main text Fig 5B. (EPS)

**S9 Fig. Posterior variance of function estimates under different models.** Each plot shows the posterior variance of a function at each time point under each of $M_{batch}$ and $M_{full}$ versus $M_{null}$. (A) $\delta(t)$ estimated for *H. salinarum* growth under low (left) and high (right) OS. (B) $(\alpha\beta)_{p,c}(t)$ at pH = 5, mM malic acid = 10. (EPS)

# Author Contributions

**Conceptualization:** Peter D. Tonner, Amy K. Schmid, Scott C. Schmidler.

**Data curation:** Peter D. Tonner, Cynthia L. Darnell, Francesca M. L. Bushell, Peter A. Lund, Amy K. Schmid.

**Formal analysis:** Peter D. Tonner, Cynthia L. Darnell, Scott C. Schmidler.

**Funding acquisition:** Peter A. Lund, Amy K. Schmid, Scott C. Schmidler.

**Investigation:** Peter D. Tonner, Cynthia L. Darnell, Francesca M. L. Bushell, Peter A. Lund, Amy K. Schmid, Scott C. Schmidler.

**Methodology:** Peter D. Tonner, Cynthia L. Darnell, Francesca M. L. Bushell, Peter A. Lund, Amy K. Schmid, Scott C. Schmidler.

**Project administration:** Peter A. Lund, Amy K. Schmid, Scott C. Schmidler.

**Resources:** Peter A. Lund, Amy K. Schmid, Scott C. Schmidler.

**Software:** Peter D. Tonner.

**Supervision:** Peter A. Lund, Amy K. Schmid, Scott C. Schmidler.

**Validation:** Peter D. Tonner, Cynthia L. Darnell, Francesca M. L. Bushell.

**Visualization:** Peter D. Tonner.

**Writing – original draft:** Peter D. Tonner.

**Writing – review & editing:** Peter D. Tonner, Cynthia L. Darnell, Francesca M. L. Bushell, Peter A. Lund, Amy K. Schmid, Scott C. Schmidler.

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
