## [Decision Letter · Decision Letter 0]

23 Jun 2020

Dear Dr. Schmid,

Thank you very much for submitting your manuscript "A Bayesian Non-parametric Mixed-Effects Model of Microbial Phenotypes" for consideration at PLOS Computational Biology.

As with all papers reviewed by the journal, your manuscript was reviewed by members of the editorial board and by several independent reviewers. In light of the reviews (below this email), we would like to invite the resubmission of a significantly-revised version that takes into account the reviewers' comments. The reviewers bring up important points, but they all seem quite addressable.

We cannot make any decision about publication until we have seen the revised manuscript and your response to the reviewers' comments. Your revised manuscript is also likely to be sent to reviewers for further evaluation.

Sincerely,

Jason Papin

Editor-in-Chief

PLOS Computational Biology

Reviewer's Responses to Questions

**Comments to the Authors:**

Reviewer #1: Summary:

The authors present a new Bayesian modeling approach for cell growth that is adaptable, allows for incorporation of several effectors, and includes batch-to-batch variation. While there is novelty to the overall approach, the manuscript is difficult to read and the implementation seems convoluted. If these issues can be addressed, the approach described could be adopted widely.

Major Points:

Overall, I found the modeling approach intriguing but the manuscript somewhat difficult to read. This is unfortunate given that growth models of microbes should be a topic accessible by a wide audience. The modeling approach described here has the potential to be adopted somewhat widely, as many engineers and biologists are looking for better models to describe microbial growth. The ability to incorporate batch-to-batch effects is certainly a benefit, and the ability to add in additional effects, inhibitors, etc. into the framework may make this approach universally adaptable. With that, there are problems that should be addressed. Currently, it is not clear how one would apply this modeling framework to build a model for monitoring bioprocesses. What phenomics experiments should be run a priori? How are model parameters found? How many batches and replicates are needed? Is batch-to-batch variability reduced by including more variables in the model? The code is available through GitHub, but the implementation in Python is not for the inexperienced user. From this, my main suggestion is to improve accessibility and provide user-friendly instructions and guidelines. This could lead to a large number of citations.

The introduction does not mention the contribution of “genome-scale” kinetic and dynamic flux-based modeling strategies in determining growth rates, effects of inhibitors, multiple substrates, genetic perturbations, etc. There is a wealth of knowledge in the literature that should be referenced. How does the Bayesian modeling approach proposed in this paper compare to the other dynamic genome-scale approaches that try to incorporate mechanistic effects?

Eq. 6 contains several conditional statements, and these can be expanded to incorporate other conditions, I’m assuming. How is this different from other parametric Monod-type models that include terms for inhibition, multiple substrates, etc.? Don’t they all contain constants that must be found through experimental data? Can’t a Monod-type model contain a batch-to-batch variability parameter? An effective approach would be to compare several different model types given the same experimental data generated in this study. There is some reference to previously published studies in this regard, but the findings remain unclear. It seems some direct comparisons can be made in this manuscript.

It’s not clear why a single batch can contain multiple growth curves. Are these all of the same medium lot? All of the curves seem to be related even though they were grown separately. A batch is usually from a single fermenter. I'm not sure how to address this better, but it is another somewhat confusing aspect in the manuscript.

Reviewer #2: This manuscript by Tonner et al. describes a much-needed contribution to the rigorous modeling of bacterial growth. They extend their previous work on Bayesian modeling of microbial growth curves by creating a fully non-parametric framework that accounts for experimental and technical variability. They demonstrate how incorporation of batch modeling terms can change interpretation of results from growth experiments performed in varying media conditions for P. aeruginosa and H. salinarum. Overall, the manuscript is very well written and logically builds on previous work. This approach will advance studies of microbial growth curves, most of which currently apply heuristic methods for summarizing growth curves into a single metric (growth rate, AUC, etc.). I have essentially no concerns related to the utility or implementation of the phenom method, but highlight one issue with the data source for P. aeruginosa that warrants reprocessing and reanalysis. I will use the methods the authors have developed in my own projects and I am very enthusiastic about the line of research being developed by the manuscript authors.

Major concerns

1. Unrealistic values of log(OD) are plotted in most figures describing growth of P. aeruginosa in the manuscript. One concrete demonstration of the problem is in the P. aeruginosa plots in Figure 1B; log(OD) = 5 is reported in many panels. Even assuming a base of 2, that is a raw OD of ~30, an order of magnitude higher than the maximum values that might be expected in very favorable growth conditions. What is the base of the logarithm? In digging through reference 23, it was reported that raw OD values were log2 transformed; however, the raw values in the github repository for that paper are between around 0-1.5 yet the processed data are in the range of around 0-6. In the paper itself, all plotted OD values seem to be from the raw data rather than processed data (or a different version of the processed data). In the publication’s github repo, it looks like there may be a problem with the scaling process (https://github.com/amyschmid/pseudomonas-organic-acids/blob/master/preprocess.ipynb).

Specifically, the problem seems to be in this loop:

for k,index in group.groups.items():

temp = data.loc[:,index]

od = temp.values[:5,:].ravel()

coeff = np.polyfit(time.tolist()*temp.shape[1],od,2)

temp = temp - np.polyval(coeff,data.index.values[0])

A second order polynomial is fit to the first 5 time points, and then the polynomial is evaluated at the first time point. This results in subtraction of a large negative number (-6.17 for the example growth curve I used), inflating the OD for the entire growth curve. The motivation for the polynomial fitting procedure on only the first five time points, then the scaling by the polynomial fit value for only the first time point, is unclear to me. Why not remove the first five time points, and scale the entire curve by subtracting by the real value of the first time point? The choice to scale based on a within-sample polynomial fit may also introduce artificial technical variation between samples.

Please resolve this issue with the data, rerun all analyses for P. aeruginosa, and include the base of the logarithm in the axis titles or the figure legend. I do not think that this issue points to any underlying problems with the phenom method. The H. salinarum data do not raise any red flags for me, but I would encourage you to inspect the data with the same rigor that you should for P. aeruginosa, given the potential errors.

2. Starting around the text describing Figure 4, “significant” starts to be used in a way that isn’t defined in the text (line 207, “significantly different posterior estimates”, line 220, “... is estimated to be significant during the time points...”; again on lines 221, 222, 224). Please define the procedure used to establish significance in these cases, or use a different term if a formal procedure was not used (which seems completely fair to me given the use of fully probabilistic models).

3. This is not a concern but a strong recommendation: the points about shrinkage of the credible interval with Mfull in Figure 5C are perhaps the most important in the manuscript, but are hard for the reader to grasp because of the overlapping intervals. The point is clear in the text but could be drawn out more clearly in the figure. There are many ways you could emphasize this, but I think the easiest would be having subpanels beneath each panel in Figure 5C which individual plots the estimates of Mnull, Mbatch, and Mfull (e.g., 3 plots with individual models and one plot with estimates from all models shown together for context).

Minor concerns

1. Lines 106-112, “technical”, “biological”, and “batch” should be defined by describing the experiments performed. Based on my own terminology usage, I assume for H. salinarum that 9 experiments were performed in which a frozen stock was struck on plates, a colony/lawn was gathered and pre-cultured, and the preculture was standardized via OD and used to inoculate each well (“9 batches”). Within each of these 9 separate experiments, each environmental condition/media condition was replicated in quadruplicate (i.e., “4 biological replicates”), and at each time point OD was measured from in each of the biological replicate wells three times (i.e., “3 technical replicates”). Many other researchers consider the “biological replicates” I describe here to be “technical replicates”. We also recommend indicating whether media and/or supplements (e.g., organic acids) were made separately for each “batch”, or whether the same solutions were added to media ingredients form the same lot, etc.

2. Line 292; “Cultures were then diluted to OD600~0.05 in a high throughput microplate reader” - was the culture diluted to an OD600 of ~0.05 as measured in a 1cm cuvette and then transferred to the microplate, or was culture diluted such that within each well of the microplate the OD600 was 0.05 at the beginning of the growth experiments as measured by the platereader? Please clarify, and if the latter is the case, please indicate the format of the microplates (e.g., 96-well flat-bottom) and the total volume used in each well. Please provide a similar level of detail in the description of culture setup for P. aeruginosa as well.

3. All figure captions and supplemental figure captions: mu and delta are occasionally expressed as a function of x (instead of t, as in the rest of the manuscript).

4. Labelling some panels of Figure 4 with species information would make the figure more interpretable; e.g., having only two batches in panel C makes it tough to figure out that the two colors are still referring to different batches and not some new concept.

5. In Figure 4C, why is the posterior estimate of the interaction term expressed as the derivative of the growth curve rather than using log(OD) as for all the other estimate plots?

6. In Figure 5A, label the left/right panels as low/high OS, respectively, on the figure itself

7. Lines 71-73, “The presence…” is incomplete, seems like the middle of the sentence should have the following inserted: “..., but [these efforts did not include modeling of] individual batch effects for each term in the model.”

Below are recommendations that we believe would improve the readability of the manuscript but do not think necessary to revise as part of the peer review process.

The broad readership of PLOS Computational Biology would benefit from explicit definition of several statistical modeling terms used throughout the paper. We recommend defining the following terms at first mention in the introduction. Some of these terms are defined in the text but could use additional clarification. The following are just a few of these terms; it may help to have an application-oriented computational biologist read your manuscript and identify terms that they don’t understand which limit their ability to understand the purpose and findings of the paper.

1. fixed effects. This definition seems suitable

2. random effects. This definition seems suitable but would benefit from using a statistical synonym for “population” since population implies something different in “population growth model”. Conversely, “population growth model” could use a biological synonym for “population”.

3. parametric model. Clearer indication that the parameters that make standard population growth models “parametric” are those that bake in the assumptions of lag-log-stationary, rather than another assumption about a distribution.

4. secondary model. The average reader will not be able to define “secondary model” based on the context provided alone.

Reviewer #3: This paper presents phenom, a method and software tool for fitting non-parametric mixed-effects models to microbial growth curves. The method is a generalized version of previous work by the same authors with a focus on handling batch and replicate effects without relying on a parametric growth curve. My main concerns are twofold: 1.) are interesting parts of the stress response swept into batch effects, and 2.) do the perturbation functions extracted by phenom correspond to the underlying biology?

Major Comments

My first concern is largely technical. The phenom method assumes that batch effects are simply random effects that perturb the true underlying \\mu(t). As a microbiologist, I'm not sure this is true. Take, for example, the PA growth curves if Fig. 1B with pH 5.5 and 10 mM acid. I have a hard time believing that the \\mu(t) underlying these curves are the same across the two batches. Bacteria do not have a single response to every combination of stressors. Small (random) perturbations early on could lead to vastly different transcriptional states and therefore two different functions \\mu(t). My interpretation of these data is the conditions (pH 5.5 and 10 mM acid) are an unstable region of the stress response network. There are actually two \\mu(t), or at least two functions \\delta(t), that the organism is pushed into randomly. Said another way, this is not a batch effect but rather a phase transition.

One way to analyze this is to look at how the function \\delta(t) behaves over the experimental region. Does phenom correctly place a steep transition around pH=5.5 and 10 mM acid? If not, then I would be concerned that the batch effects are hiding large changes in the stress response.

My second, and most significant, point is that the presentation of phenom feels disconnected from the underlying biology. As the authors mention, growth curves are rich sources of data regarding phenotypes and stress responses. The main claim of phenom is that it can dissect different sources of variability to uncover the biologically important signal. The paper shows many examples where batch effects supposedly hide perturbations, but there is no ground truth by which we can assess if the uncovered perturbation functions are correct. Many of these data were collected from other papers that study the underlying biology. It is important to show that the differences between the improved models more accurately reflect the stress response.

For example, the authors note that M_null finds significant perturbations from 10-40 hours, but M_batch reports the significant interval is 20-40 hours (line 220). Which is correct? Without comparing these intervals to the biological stress response, we cannot say which method is better; we can only say that they are different.

Ultimately, phenom will only be useful if it can provide insight into the underlying biology. I suggest the authors place a greater emphasis on how well their method translates OD measurements into latent biology. It would be fantastic if the authors can show how phenom could reduce the need for detailed intracellular characterization of the stress response, or at least identify interesting timepoints for further study.

Minor Comments

I thought the paper was very well written. It was clear and concise.

I suggest changing the title from "microbial phenotypes" to "microbial growth curves". "Phenotypes" is a broad term that includes almost any measurable feature of a microbe, e.g. toxin secretion or sporulation. This paper focuses on only one phenotype, i.e. growth. Growth is a very important phenotype, so I don't believe the more precise title diminishes the work.

The notation can be confusing, although I believe the confusion stems from notational conventions in two different fields rather than poor choices by the authors. The variable \\mu is used in microbiology as the instantaneous growth rate of a microbe. The authors use this convention when describing their parametric models (\\mu_{max}, line 126). Later (line 148) the authors define \\mu(t) as the "average growth behavior of an organism". From Figure 3 and eq (1), we see that this definition of \\mu(t) implies the units of \\mu are population size or [OD]. My suggestion is to change \\mu(t) to some other name. This may be strange for statisticians, but I believe research focused solely on microbial growth curves should not repurpose the standard names for the parameters in a sigmoid growth curve.

**Have all data underlying the figures and results presented in the manuscript been provided?**

Reviewer #1: Yes

Reviewer #2: Yes

Reviewer #3: Yes

PLOS authors have the option to publish the peer review history of their article (what does this mean?). If published, this will include your full peer review and any attached files.

Reviewer #1: No

Reviewer #2: **Yes: **Greg Medlock

Reviewer #3: No
---

## [Decision Letter · Decision Letter 1]

30 Aug 2020

Dear Dr. Schmid,

We are pleased to inform you that your manuscript 'A Bayesian Non-parametric Mixed-Effects Model of Microbial Growth Curves' has been provisionally accepted for publication in PLOS Computational Biology.

Best regards,

Jason A. Papin

Editor-in-Chief

PLOS Computational Biology

Jason Papin

Editor-in-Chief

PLOS Computational Biology

Reviewer's Responses to Questions

**Comments to the Authors:**

Reviewer #1: The authors have addressed my concerns regarding their manuscript. I encourage the authors to continue improving the accessibility and implementation of their method through shared computational resources in GitHub. The method must be easy to implement for microbiologists with minimal programming expertise. This will result in considerably more citations for this article, in my opinion.

Reviewer #2: The authors have thoroughly addressed all of my comments.

Reviewer #3: The authors have sufficiently addressed my concerns. The revised paper more accurately describes phenom as a statistical tool, and less-so as a method for understanding biology. I think it will be a useful contribution to the field.

**Have all data underlying the figures and results presented in the manuscript been provided?**

Reviewer #1: Yes

Reviewer #2: Yes

Reviewer #3: Yes

PLOS authors have the option to publish the peer review history of their article (what does this mean?). If published, this will include your full peer review and any attached files.

Reviewer #1: No

Reviewer #2: **Yes: **Greg Medlock

Reviewer #3: No

---

## [Editor Report · Acceptance letter]

19 Oct 2020

PCOMPBIOL-D-20-00079R1 

A Bayesian Non-parametric Mixed-Effects Model of Microbial Growth Curves

Dear Dr Schmid,

I am pleased to inform you that your manuscript has been formally accepted for publication in PLOS Computational Biology. Your manuscript is now with our production department and you will be notified of the publication date in due course.

With kind regards,

Matt Lyles
